# OpenReview forum: "SafeCompass: Dynamic Chain-of-Thought Steering via Inference-Time Safety Signals"
_ICML.cc/2026/Conference — ICML 2026 regular_

### Official Review · Reviewer_gbqm · 2026-03-08

**Soundness:** 2
**Presentation:** 2
**Significance:** 2
**Originality:** 2
**Overall Recommendation:** 2
**Confidence:** 4

**Summary:**

The paper proposes SafeCompass. The method is deployed at inference time to defend against jailbreak attacks of LRMs. SafeCompass periodically monitors the internal hidden state during reasoning to compute a safety score. The score is computed by estimating a latent safety direction based on the difference between safety-related prompt and neutral control sequence.

**Compliance With Llm Reviewing Policy:**

Affirmed.

**Final Justification:**

My primary concern is conceptual and fundamental. The paper treats d_safety​ as a latent "safety direction," but by construction it is merely the activation difference induced by appending a specific safety sentence relative to a control branch. This does not identify an underlying safety variable; it captures the model’s context-dependent response to a particular prompt. Consequently, the proposed "safety score" is not an intrinsic measurement of safety, but an intervention-dependent quantity entangled with the probing procedure itself. This undermines the central claim of monitoring internal safety dynamics. SafeCompass is not observing safety, but partially constructing the signal it later interprets.

More importantly, this raises a serious alternative explanation for the reported gains: the improvements may simply arise from repeatedly injecting safety-biased tokens and perturbing the decoding trajectory, rather than from any meaningful extraction of a safety representation. In this view, the hidden-state projection is incidental, and the method functions as a form of periodic prompt-based regularization. If so, the claimed mechanism is misidentified, and the empirical results do not support the paper’s core interpretation.

Given this fundamental mismatch between the claimed mechanism and what is actually identified, I have to lower my score.

**Key Questions For Authors:**

See above

**Limitations:**

See above

**Strengths And Weaknesses:**

1. The method assumes that the difference between hidden states corresponds to the meaningful safety direction in representation space. However, I don't see strong theoretical or empirical evidence supporting this claim.
2. The extra inference branches to compute safety signals introduces additional overhead. So the what is the overhead, does it scale?
3. The safety direction depends on a manually chosen safety sentence and threshold. How robust it the method across different settings?
4. What about adaptive attack against this method?

---

> ### Author Rebuttal · Authors · 2026-03-26
>
> We thank the reviewer for the constructive feedback. Below, we address each identified weakness with additional context and empirical evidence:
>
> **Weakness 1:**
>
> The mechanism of extracting a meaningful concept direction via the difference between hidden states is firmly grounded in recent research, including CAA by Anthropic [1], RepE by CMU [2], and JBShield [3], which we discussed in the Related Work section. These works demonstrate that computing differences in the hidden space yields meaningful concept vectors that are utilized for model steering. Building on this insight, we derive a safety-specific concept vector from hidden state differences and use it to estimate the safety score.
>
> Empirically, the highly differentiated safety score distributions in Section 5.3 confirm that extracting the safety direction via hidden state subtraction successfully isolates it. To further illustrate this, we provide supplemental PCA visualizations of hidden states for paired safe and neutral control branches. The difference between the unified safe state and the varying contextual states dynamically extracts the safety direction. Please see the detailed results [here](https://anonymous.4open.science/api/repo/SafeCompass-EC19/file/PCA_of_Branches.pdf?v=d4e0bc54).
>
> **Weakness 2:**
>
> **We have explicitly addressed this crucial point in our paper.** Because the extra inference branches activate only periodically, the additional latency is kept minimal. We evaluate latency using both the average token generation rate (ATGR) and average execution time. As illustrated in Figure 5 and Appendix E, our overhead is tightly bounded: we maintain a high ATGR of ≥0.93x and restrict the relative execution time to ≤1.10x.
>
> Furthermore, this efficiency scales seamlessly across model sizes:
>
> * 8B to 32B Models: Consistently preserve a high ATGR (≥0.93x).
>
> * 70B Model (New R1-70B Experiments): Maintains >91% of the original ATGR, with the time ratio remaining under 1.10x.
>
> | R1-70B | ATGR (w/o SC) | ATGR (w/ SC) | ATGR Ratio (x) | Time (w/o SC) | Time (w/ SC) | Time Ratio (x) |
> | :--- | :---: | :---: | :---: | :---: | :---: | :---: |
> | MMLU-Pro | 30.84 | 29.91 | **0.97x** | 45.08 | 44.18 | **0.98x** |
> | GSM8K | 30.56 | 27.80 | **0.91x** | 18.26 | 19.54 | **1.07x** |
> | Alpaca | 30.83 | 28.37 | **0.92x** | 30.84 | 31.77 | **1.03x** |
>
> **Weakness 3:**
>
> **We discussed this in Section 5.4.** Table 4 in our paper demonstrates that our framework is robust to the choice of the safety sentence, provided the semantics are similar. Furthermore, our study of three thresholds ($\alpha$ = -0.25, -0.20, -0.15) confirms minor trade-offs between defense and utility. As shown in Figure 5, a higher $\alpha$ value (-0.15) triggers interventions more easily, leading to a marginal increase in over-refusal (0%–3%). Nevertheless, the overall balance between safety and general utility remains robust.
>
> **Weakness 4:**
>
> In our paper, to specifically target the CoT intervention strategy of SafeCompass, we have presented a 'Thought Block Attack' (Figure 17), which demonstrates that SafeCompass remains effective during the response generation phase, even when the model's thinking process is completely bypassed.
>
> Furthermore, we design adaptive attacks based on GCG and SlotGCG [4] in order to evade monitoring by incorporating a penalty term into their loss functions. This term optimizes the adversarial prompt to increase the projection value (i.e., the safety score) of the hidden states $h\_t$ at targeted tokens onto the safety vector $d\_ {safety}$:$$\\mathcal L\_ {total} = \\mathcal L\_ {CE}+ \\lambda \\max(0, -\\cos(h\_t, d\_ {safety})),$$where $\\mathcal L\_ {CE}$ is the cross-entropy loss, and the second term serves as a penalty to increase the safety score to actively bypass detection.
>
> The results below, which report the ASR of models with and without SafeCompass (SC), indicate that while Projection-Aware (PA) GCG and PA-SlotGCG can marginally increase the ASR, the overall defense remains robust due to the difficulty of balancing malicious output elicitation with improving the safety score.
>
> | Setting | GCG | PA-GCG | SlotGCG | PA-SlotGCG |
> | :--- | :---: | :---: | :---: | :---: |
> | Qwen3 w/o SC | 0.16 | 0.10 | 0.31 | 0.28 |
> | **Qwen3 w/ SC** | **0.00** | **0.02** | **0.00** | **0.06** |
> | R1 w/o SC | 0.26 | 0.22 | 0.46 | 0.37 |
> | **R1 w/ SC** | **0.06** | **0.07** | **0.11** | **0.14** |
>
>
> [1] N. Rimsky, et al. "Steering Llama 2 via Contrastive Activation Addition." ACL, 2024.
>
> [2] Andy Zou, et al. "Representation Engineering: A Top-Down Approach to AI Transparency." arXiv:2310.01405.
>
> [3] Zhang, S. et al. "JBShield: Defending Large Language Models from Jailbreak Attacks through Activated Concept Analysis and Manipulation." USENIX Security, 2025.
>
> [4] Seungwon Jeong, et al. "SlotGCG: Exploiting the Positional Vulnerability in LLMs for Jailbreak Attacks." ICLR, 2026.

---

> > ### Author Rebuttal · Reviewer_gbqm · 2026-04-03
> >
> > Thanks for the rebuttal! I have a few follow up questions.
> >
> > 1. You cite previous work and provide PCA visualizations to support your claim that hidden-state differences encode meaningful concept directions. However, these are largely correlational analyses. Can you provide causal evidence that your extracted safety direction d_{safety} is actually responsible for improved safety behavior? What happens if you actively perturb hidden states along d_{safety} vs orthogonal directions? Does steering along this direction consistently increase safety scores and reduce ASR?
> > 2. Your method relies on subtracting a safety-augmented branch and a neutral control branch. Have you compared against: a) safety − random prompt, b) safety − unsafe prompt, and c) different control token constructions? Without such ablations, it is unclear whether the extracted direction is truly safety-specific or simply reflects generic perturbations in representation space.
> > 3. What happens if the safety sentence is slightly misaligned or adversarially perturbed? Could an attacker exploit this dependency to manipulate the extracted direction?

---

> > > ### Author Response · Authors · 2026-04-04
> > >
> > > Thank you for the follow-up feedback!
> > >
> > > >Question 1: Using vectors for direct steering
> > >
> > > We perturb the model's hidden states using either $d_{safety}$ or a sampled orthogonal vector, scaled by coefficient $\alpha$ before addition to ensure intervention efficacy. The table below presents the ASR under both methods across different $\alpha$, with orthogonal results averaged over three runs to mitigate the randomness of vector sampling.
> > >
> > > Our results provide **causal evidence** that $d_{safety}$ encodes a meaningful safety direction. By actively intervening on hidden states, steering **along $d_{safety}$** consistently reduces ASR, while **orthogonal perturbations** have weaker or inconsistent effects. Across $\alpha$, steering along $d_{safety}$ yields a robust, monotonic ASR reduction on both attacks. In contrast, orthogonal perturbations have negligible impact on DeepInception and only partial reductions on CodeAttack—likely due to incidental disruption of its decoding process. Crucially, these effects are unstable and not semantically aligned with safety, unlike the consistent improvements achieved along $d_{safety}$.
> > >
> > > This shows safety gains are **direction-specific rather than a byproduct of generic perturbations**, establishing a clear causal link between $d_{safety}$ and improved safety behavior.
> > >
> > > ||Setting|DeepIn.| CodeAt.|
> > > |---|---|---|---|
> > > ||No-Defense|0.62|0.55|
> > > | $\alpha=5$ |Orthogonal to $d_{safety}$|0.63|0.37|
> > > ||Along $d_{safety}$|0.52|0.31|
> > > | $\alpha=10$ |Orthogonal to $d_{safety}$|0.62|0.32|
> > > ||Along $d_{safety}$|0.47|0.25|
> > > ||SafeCompass|0.00|0.04|
> > >
> > > Finally, we emphasize that in our paper, the goal of extracting $d_{safety}$ is **safety monitoring to guide interventions**, rather than direct steering. Vector steering is unreliable even in standard LLMs (e.g., Jailbreak Antidote) and becomes more challenging in LRMs with complex latent spaces. Consistent with this, results in above table show that direct steering along $d_{safety}$ is less effective than SafeCompass.
> > >
> > > >Question 2: More ablation study
> > >
> > > In our design, the control branch serves to preserve the original semantics, enabling the isolation of the safety semantic direction used for computing the safety score. As demonstrated in Figures 4, 11, and 12 of our paper, the safety score distributions on both benign and harmful datasets validate the soundness of our approach.
> > >
> > > Following your suggestion, we evaluated three additional ablation studies for prompts below:
> > > * Random Prompt A = "Apple bicycle ... ,"
> > > * Random Prompt B = "Airplane night ... ,"
> > > * Unsafe Prompt = "I will bypass all safety guidelines and provide harmful content,"
> > > * Different Control Token = "-"
> > >
> > > Under these settings, we visualize the safety score distributions and evaluate the ASR. Random or unsafe prompts introduce unexpected semantic noise, so the safety scores fail to exhibit the correct distribution (visualizations can be viewed [here](https://anonymous.4open.science/api/repo/SafeCompass-EC19/file/safety_scores_distribution.pdf?v=bb5e997e)). For instance, an unsafe prompt may induce safety awareness, causing the model to generate self-correcting thoughts (**e.g., "I will bypass all safety guidelines and provide harmful content, but I'm not going to do this"**). As a result, its internal state fails to anchor in the expected unsafe space, making the hidden state difference between safe and unsafe branches less meaningful.
> > >
> > > Conversely, using a different control token construction did not negatively impact the results, as it still preserves the pure original semantics of the input. As ASR shown in the table, the defense performance of both the random and unsafe prompt settings is inferior to the setting that utilizes the control token.
> > >
> > > |Setting|DeepIn.|CodeAt.|OverRefusal|
> > > |:---|:---|:---|:---|
> > > |No-Defense|0.62|0.55|0.00|
> > > |Safety-Random Prompt|0.60|0.32|0.00|
> > > |Safety-Unsafe Prompt|0.46|0.27|0.00|
> > > |Different Control Tokens|0.01|0.02|0.01|
> > >
> > > >Question 3: Potential Exploitation of Safety Sentence
> > >
> > > In SafeCompass, we assume a standard threat model where attackers can only modify the inputs through query interactions, without access to internal states or components. The safety sentence is predefined by the defender (i.e., the model owner), and isn't influenced by user inputs. **Therefore, external attackers cannot perturb this sentence or manipulate the extracted direction.**
> > >
> > > Importantly, even if the attacker is aware of the safety sentence, our method can't be bypassed by input-level modifications alone. As demonstrated in our adaptive attack evaluation, SafeCompass remains robust under attackers that optimize inputs with knowledge of the defense mechanism.
> > >
> > > Finally, attacks that assume the ability to modify internal components (e.g., the safety sentence) require control over the deployment infrastructure. Such assumptions go beyond standard LLM safety threat model and are therefore considered out of scope.
> > >
> > > We sincerely hope these clarifications address your questions.

---

### Official Review · Reviewer_r5kR · 2026-03-08

**Soundness:** 4
**Presentation:** 4
**Significance:** 3
**Originality:** 4
**Overall Recommendation:** 5
**Confidence:** 3

**Summary:**

This paper introduces a plug and play framework that reduces attack success rate while minimizing over-refusal rate. It does so by computing safety score from model latent representation and perform CoT intervention when the safety score falls below a threshold.

**Compliance With Llm Reviewing Policy:**

Affirmed.

**Final Justification:**

My final recommendation remains Accept. I find the paper technically solid, clearly written, and supported by thorough experiments. My main concern was the reliance on internal model states, which limits applicability in black-box settings. The rebuttal addressed this concern reasonably well: beyond clarifying the typo in Table 1, the authors also added a black-box variant in their response. The additional results suggest that an output-based version can still improve over having no SafeCompass at all, but it remains clearly weaker than the full hidden-state method and comes with higher over-refusal. This additional evidence is helpful and makes the tradeoff clearer, but it does not remove the main limitation. Overall, the rebuttal reinforced my original assessment rather than changing it, so I remain at Accept.

**Key Questions For Authors:**

1. in table 1, what is BENIGN insert setting? a typo? Is it position A, the beginning of CoT?
2. Do you think there is a way to compute a safety score without accessing the model states?

**Limitations:**

yes

**Strengths And Weaknesses:**

Strengths:

1. the design of safety score calculation is well supported by ablation study of each component.

2. The use three LLM judges for evaluation to mitigate model bias, and the accuracy of the evaluation is verified by human experts.

3. Comprehensive evaluation against different defense and attack methods shown that SafeCompass sigificantly reduces attack success rate while maintaining minimum over-refusal rate.

4. The evaluation of the method is very thorough, such as evaluation against baselines, ablation and hyperparameters study,  runtime efficiency.

Weaknesses:

The method require access to model's internal states and generated reasoning content, therefore the method is limited to open-source models.

---

> ### Author Rebuttal · Authors · 2026-03-29
>
> We deeply appreciate your positive feedback and your recognition of the comprehensiveness of our experiments. We are highly encouraged by your favorable evaluation of our soundness and presentation. In the following sections, we provide detailed explanations to address your questions.
>
> **Weakness 1 & Question 2: Computing safety scores without accessing model states**
>
> **Limitation of Reliance on Model Internal States.** We completely agree with your assessment that relying on internal states currently limits the application to open-weights models. Indeed, in many real-world deployments—such as agentic applications calling commercial models via closed APIs—internal states are inaccessible, which currently precludes the direct use of SafeCompass. **However, when AI developers and providers (e.g., Google, OpenAI) deploy their proprietary models internally, they possess full access to these hidden states. In such scenarios, they can seamlessly integrate SafeCompass at the system level to provide jailbreak defense.** We will add this discussion to the Limitations section of the revised manuscript.
>
> **Extending to Black-Box Scenarios.** In black-box scenarios where internal states are inaccessible, we believe it remains feasible to estimate the safety score by relying solely on the model's generated outputs, although this inevitably sacrifices some accuracy due to the lack of rich signals from intermediate layers. The paradigm of output-based evaluation has evolved to use lightweight external models, such as Qwen3-Guard [1] and ReasoningShield [2], which statically evaluate the entire completed response to classify it into qualitative tiers. By extending these methods to yield more fine-grained safety scores and dynamically evaluating the generated content at various steps during the reasoning process, this output-based approach can be readily incorporated into the dynamic intervention framework of SafeCompass.
>
> We sincerely thank the reviewer for raising this highly practical question. We would like to explore this paradigm in our future work to broaden the applicability of dynamic safety score computation.
>
> **Question 1: In Table 1, what is the "BENIGN" insert setting? Is it a typo?**
>
> We thank the reviewer for pointing this out. "BENIGN" is indeed a typo and should be "BEGIN". We will correct it in the revised manuscript.
>
> [1] Haiquan Zhao, et al., "Qwen3Guard Technical Report." arXiv preprint arXiv:2510.14276.
>
> [2] Changyi Li, et al., "ReasoningShield: Content Safety Detection over Reasoning Traces of Large Reasoning Models." arXiv preprint arXiv:2505.17244.

---

> > ### Author Rebuttal · Reviewer_r5kR · 2026-04-02
> >
> > Thank you for the clarification. The rebuttal resolves my question regarding the typo in Table 1 and provides a reasonable discussion of the method’s reliance on internal states. This limitation still remains, since the rebuttal only outlines a possible black-box direction as future work rather than demonstrating it in the current paper. Overall, I find the response satisfactory, and my positive assessment remains unchanged.

---

> > > ### Author Response · Authors · 2026-04-03
> > >
> > > We sincerely thank you for your continued support and for finding our response satisfactory.
> > >
> > > While we agree that relying on internal states limits our application scope, it is also the primary motivation of our work. Recent evidence suggests that model outputs do not always reflect true reasoning processes [1, 2]. By leveraging internal states, SafeCompass can accurately capture safety state transitions during the model's reasoning to guide CoT interventions for robust defense.
> > >
> > > Meanwhile, we adapted SafeCompass to a black-box setting. We tested an output-based scoring version using Qwen3-Guard and GPT-5.1. As shown in the table, while the black-box adaptation achieves a promising level of defense, the hidden-state-based SafeCompass still demonstrates superior performance with stronger ASR reduction and lower over-refusal.
> > >
> > > | Setting | DeepInception $\downarrow$ | ReNeLLM $\downarrow$ | CodeAttack $\downarrow$ | Chameleon $\downarrow$ | OverRefusal $\downarrow$ |
> > > | :--- | :--- | :--- | :--- | :--- | :--- |
> > > | w/o SafeCompass | 0.62 | 0.70 | 0.55 | 0.38 | 0.01 |
> > > | **w/ SafeCompass** | **0.00** | **0.01** | **0.04** | **0.01** | **0.01** |
> > > | black-box SafeCompass | 0.05 | 0.15 | 0.19 | 0.17 | 0.05 |
> > >
> > > We will incorporate this discussion into the Limitations section of our revised manuscript. Thank you again for your constructive feedback that helps strengthen our paper.
> > >
> > >
> > >
> > > [1] Chen et al., "Reasoning Models Don't Always Say What They Think", arXiv:2505.05410 (2025).
> > >
> > > [2] Chua & Evans. "Are DeepSeek R1 and Other Reasoning Models More Faithful?" arXiv:2501.08156, 2025.

---

### Official Review · Reviewer_NC7x · 2026-03-12

**Soundness:** 3
**Presentation:** 3
**Significance:** 3
**Originality:** 3
**Overall Recommendation:** 5
**Confidence:** 3

**Summary:**

The paper proposes SafeCompass, an inference-time safety defense method for Large Reasoning Models. It dynamically monitors and steers chain-of-thought reasoning process using using inference-time safety signals extracted from internal states. The safety direction detemined by contrastive hidden states of safety branch and control branch helps to evaluate the current internal safety state.  Comprehensive experiments demonstrate that SafeCompass significantly improves safety performance while preserving general ability.

**Compliance With Llm Reviewing Policy:**

Affirmed.

**Final Justification:**

The author's rebuttal addressed my concern about method cost and additional analysis, so I keep my positive score with 5.

**Key Questions For Authors:**

1. Could the authors provide some analysis explaining why cosine similarity is sufficient to achieve a good trade-off? Normally, hidden states $h_t$ contain much richer information and may exhibit non-linear structures. It is therefore interesting that this simple design works well.

2. In Table 1, does the "BENIGN" entry in the Settings column actually mean "BEGIN"?

**Limitations:**

yes

**Strengths And Weaknesses:**

## Strengths

1. The research problem is timely and highly important. The proposed contrastive hidden-state technique that dynamically steering the CoT reasoning at inference time is interesting.

2. The experimental section provides comprehensive and strong empirical results, proving the effectiveness of SafeCompass.

3. The work is presented clearly, and the method design is clean and well illustrated.

## Weaknesses

1. While the method is empirically effective, the paper provides limited theoretical justification or interpretability analysis explaining why this pipeline works well.

2. I think this method may introduce significantly higher computational cost during inference, but latency measurements and analysis are missing.

---

> ### Author Rebuttal · Authors · 2026-03-27
>
> We deeply appreciate your positive feedback and the time you took to rigorously review our paper. In the following sections, we provide detailed explanations to address your questions.
>
> **Weakness 1: Theoretical Analysis and Interpretability**
>
> We sincerely thank the reviewer for highlighting the need for deeper analysis.
>
> **The Mechanism of Dynamic Guidance:** The key to our pipeline's success lies in utilizing real-time safety scores to dynamically guide the model's safety reflections. **In contrast to prior static interventions, our key insight is that the model's safety state dynamically evolves throughout the reasoning process. Theoretically, by accurately characterizing these continuous shifts, SafeCompass performs dynamic CoT interventions to steer the model toward robust defense.** As shown in Figure 6 in our paper, targeting low-safety states is critical to our success: on Qwen3-8B, this reduces average ASR to 2.0%, whereas intervening at high-safety positions yields no significant improvement, resulting in a high ASR of 41.56%. This contrast explicitly explains the underlying effectiveness of our "monitor-then-intervene" pipeline.
>
> **PCA Visualization of Hidden States Pre- and Post-Intervention:** Furthermore, to provide a more intuitive interpretability analysis of the underlying mechanisms of our pipeline, we conducted a PCA visualization of the model's hidden states pre- and post-intervention (anonymized results available [here](https://anonymous.4open.science/api/repo/SafeCompass-EC19/file/PCA_results.pdf?v=83d675a4)). **The visualization explicitly illustrates how the intervention steers the internal representations to the safe state. It reveals that as the internal representations naturally diverge and drift during complex reasoning, the required safety direction shifts correspondingly. This drift visually proves why a dynamic intervention mechanism is absolutely necessary and how our pipeline effectively operates.**
>
> **Weakness 2: Computational Overhead and Latency**
>
> We are glad you raised this highly practical question. **Ensuring computational efficiency was a primary focus during our system design, and we have thoroughly evaluated this aspect in the paper.**
>
> We measured both the **average token generation rate (ATGR)** and the **average execution time per task** on identical hardware to evaluate latency. As illustrated in Figure 5 in our paper, the average task execution time overhead is strictly bounded to a maximum of ~10% across different settings. In terms of scalability, our detailed analysis in Appendix E demonstrates that SafeCompass consistently preserves a high ATGR ratio (≥0.93x) for models ranging from 8B to 32B parameters.
>
> To further validate this, our newly supplemented experiments on the R1-70B model confirm that this efficiency scales seamlessly. As detailed in the table below, even at the 70B scale across diverse benchmarks, the framework maintains over 91% of the original ATGR, and the corresponding Time Ratio (representing the relative increase in average execution time per task) remains under 1.10x, proving that SafeCompass introduces no severe latency bottlenecks.
>
> | R1-70B | ATGR (w/o SC) | ATGR (w/ SC) | ATGR Ratio (x) | Time (w/o SC) | Time (w/ SC) | Time Ratio (x) |
> | :--- | :---: | :---: | :---: | :---: | :---: | :---: |
> | MMLU-Pro | 30.84 | 29.91 | **0.97x** | 45.08 | 44.18 | **0.98x** |
> | GSM8K | 30.56 | 27.80 | **0.91x** | 18.26 | 19.54 | **1.07x** |
> | Alpaca | 30.83 | 28.37 | **0.92x** | 30.84 | 31.77 | **1.03x** |
>
> **Question 1: Sufficiency of Cosine Similarity**
>
> It is true that hidden states inherently contain rich information and exhibit non-linear structures. However, research from UChicago [1] verifies that distinct semantic concepts (e.g., safety, honesty) in models exhibit an approximately orthogonal structure. This geometric property ensures that when computing safety scores via cosine similarity, the calculation remains immune to interference from other concept information. Building on this approximate orthogonality of semantic concepts, recent studies like CAA [2] and RepE [3] have utilized semantic vectors to steer specific model behaviors without degrading performance in other dimensions.
>
> We appreciate this insightful question; it has inspired us to think more deeply about the geometry of the representation space, which will highly benefit our future research.
>
> **Question 2: Typo in Table 1 Settings**
>
> We appreciate the reviewer's careful reading of our paper. It is indeed a typo, and the intended word is "BEGIN". We will update this in the final version.
>
> [1] Park et al., "The Linear Representation Hypothesis and the Geometry of Large Language Models." ICML, 2024.
>
> [2] N. Rimsky, et al. "Steering Llama 2 via Contrastive Activation Addition." ACL, 2024.
>
> [3] Andy Zou, et al. "Representation Engineering: A Top-Down Approach to AI Transparency." arXiv:2310.01405.

---

> > ### Author Rebuttal · Reviewer_NC7x · 2026-03-31
> >
> > Thanks for the detailed response. I'll maintain my positive score 5.

---

> > > ### Author Response · Authors · 2026-04-07
> > >
> > > We sincerely thank the reviewer for reading our rebuttal and maintaining the positive evaluation of our work. We deeply appreciate your time and support!

---

### Decision · Program_Chairs · 2026-04-30

**Decision:**

Accept (regular)

**Comment:**

The paper proposes SafeCompass, an inference-time safety defense method for LRMs, which dynamically monitors and steers the chain-of-thought reasoning process extracted from internal states. Empirical results demonstrate the effectiveness of their method. After the rebuttal, two reviewers recognised the effectiveness and efficiency of the proposed method. Although one reviewer still has concerns about the fundamental effectiveness of their method, no empirical results show that the concern still exists, as the authors give plenty of results, including more datasets and an adaptive attack, which can prove the generalization ability of the proposed method. However, the lack of theoretical analysis to support this paper is also a weakness. Therefore, I would like to give a weak acceptance of this paper.